# Risk of Drop-Out from Follow-Up Evaluations for Celiac Disease: Is It Similar for All Patients?

**DOI:** 10.3390/nu14061223

**Published:** 2022-03-14

**Authors:** Francesco Tovoli, Chiara Faggiano, Alberto Raiteri, Alice Giamperoli, Teresa Catenaro, Vito Sansone, Dante Pio Pallotta, Alessandro Granito

**Affiliations:** 1Division of Internal Medicine, Hepatobiliary and Immunoallergic Diseases, S. Orsola-Malpighi Hospital, IRCCS Azienda Ospedaliero-Universitaria di Bologna, 40138 Bologna, Italy; francesco.tovoli2@unibo.it (F.T.); chiara.faggiano@aosp.bo.it (C.F.); 2Department of Medical and Surgical Sciences, University of Bologna, 40138 Bologna, Italy; alberto.raiteri@studio.unibo.it (A.R.); alice.giamperoli@studio.unibo.it (A.G.); teresa.catenaro@studio.unibo.it (T.C.); vito.sansone@studio.unibo.it (V.S.); dantepio.pallotta@studio.unibo.it (D.P.P.)

**Keywords:** celiac disease, compliance, complications, follow-up, gluten-free diet

## Abstract

Background: Celiac disease (CD) follow-up is a relatively underevaluated topic. However, correct adherence to follow-up procedures is central to the early recognition of complicated CD and other conditions typically associated with CD. Establishing whether patients at increased risk of complications follow clinicians’ recommendations has multiple repercussions. Methods: We retrospectively analyzed the records of patients consecutively diagnosed with CD in our outpatient clinic between January 2004 and October 2017 to investigate the factors associated with drop-out from follow-up procedures. Results: Among the 578 patients analyzed, 40 (6.9%) dropped out during the first six months and 272 (50.6%) during the observation period. The median time to drop-out was 7.4 years (95% confidence interval: 6.8–8.0). No factors were associated with early drop-out. Instead, age at diagnosis >40 years (40–59 years, *p* < 0.001; ≥60 years, *p* = 0.048) and classical clinical presentation (*p* = 0.016) were significantly associated with a lower risk of later drop-out. Conclusions: Patients at increased risk of complicated CD are more compliant with follow-up procedures than patients at lower risk, despite being prescribed the same controls. These results indirectly support the hypothesis of tailored follow-up strategies, differentiated according to the risk of complications.

## 1. Introduction

Celiac disease (CD) is a chronic condition affecting about 1% of the general population [1]. In recent years, the widespread availability of sensitive and specific serological markers of celiac disease has allowed the creation of accurate diagnostic protocols, significantly reducing the risk of both under- and overdiagnosis [2]. Once diagnosed, CD requires a lifelong gluten-free diet (GFD) to relieve gluten-related symptoms and prevent its neoplastic and non-neoplastic complications [3]. Notwithstanding the relevant improvements in gluten-free restoration, it has been reported that a GFD might have a significant socioeconomic impact [4,5,6].

Moreover, metabolic syndrome and hepatic steatosis may appear in a significant proportion of patients as early as two years after beginning a GFD [7,8,9,10]. As a consequence of these possible difficulties, all guidelines for CD provide physicians and other healthcare professionals with follow-up recommendations for CD patients [11,12,13,14,15]. These follow-up programs are not merely limited to verifying GFD adherence. Instead, they encompass a wide range of objectives, such as investigating the possible persistence of gluten-related symptoms, development of concurrent autoimmune diseases, and early detection of complicated CD [16].

Follow-up of CD patients is a relatively underevaluated topic. The evaluation of adherence to follow-up procedures as a whole offers substantial advantages in comparison with the mere evaluation of GFD adherence since follow-up evaluations also serve other pivotal scopes in the management of CD patients, such as early detection of possible cases of complicated CD. While GFD adherence plays a crucial role in preventing complications, they could arise even after the start of a well-conducted GFD, especially in patients with a late diagnosis and/or a classical presentation (i.e., with signs and symptoms of malabsorption according to the Oslo definitions) [1,17]. Currently, the suggested protocols call for a first examination six months after the beginning of a GFD and every 18–24 thereafter, regardless of patient clinical characteristics at diagnosis (“one size fits all” model) [11,12,13,14,15]. Some authors have suggested follow-up protocols differentiated according to the risk of complications, including refractory CD, ulcerative jejunoileitis, hyposplenism, lymphoma, and small-bowel carcinoma [18]. Complicated CD is, in fact, more frequent in patients diagnosed in old age and with a classical presentation (i.e., diarrhea, malabsorption syndrome). In particular, Biagi et al. [17] reported that patients older than 60 years at diagnosis of CD had an 18-fold risk of complication compared with patients diagnosed at 18–40 years and a 9 times higher risk than patients diagnosed at 40–60 years. Classical presentation also increased the risk of complications by 7 times compared to nonclassical presentation [17]. Therefore, a stricter follow-up protocol has been advocated for these patients and a more flexible one for patients without risk factors (which represent most of the incident cases) [17,18]. However, this evidence-based proposal has not yet been considered in the current guidelines for CD.

In this context, verifying whether patients belonging to these high-risk groups are compliant with follow-up procedures is especially interesting. Exploring and understanding this phenomenon is of paramount importance, as the theoretical benefit of follow-up procedures can be dissipated by reduced compliance with the procedures.

The study of the adherence of patients to the recommendations provided by healthcare professionals can be complex in countries in which CD patients are diagnosed and followed up by a wide range of physicians, including general practitioners, nutritionists, and various specialists working in first- and second-level hospital centers. As a matter of fact, data can be challenging to gather even for a mere evaluation of GFD adherence [19,20,21,22].

The Italian clinical setting, however, is characterized by some specificities. Until a few years ago, diagnosis of CD had to be confirmed by physicians working in regional referral centers to grant patients the possibility of obtaining a refund for gluten-free products approximately worth EUR 100 monthly. A large Italian study showed that economic resources allocated monthly to patients are quite adequate, thus suggesting that they do not impact compliance with GFD [23].

At the same time, most of these centers had a policy to follow-up patients indefinitely and in collaboration with their general practitioners, rather than merely referring them back after a fixed time. In the last few years, hospital teams consisting of gastroenterologists, pathologists, nutritionists, and rheumatologists have been created to improve the reliability of the diagnostic process and improve the management of follow-up problems, contributing to maintaining an approach based on large hospital centers. In such a setting, we aimed to identify the factors that can influence and affect the short and long-term adherence of patients to follow-up evaluations prescribed by their physicians according to the current guidelines.

## 2. Materials and Methods

### 2.1. Clinical Setting

We retrospectively analyzed the medical records of patients who were consecutively diagnosed with CD in our outpatient clinic (Bologna Authority Hospital S. Orsola-Malpighi, Bologna, Italy) between January 2004 and October 2016 (the final cut-off date was chosen to allow a theoretical minimum three-year follow-up even for the most recently diagnosed patients). The database was locked in February 2020, prior to the lockdown imposed by the local authorities for the COVID-19 pandemic.

### 2.2. Inclusion and Exclusion Criteria

A diagnosis of CD performed according to the North American Society for Pediatric Gastroenterology, Hepatology, and Nutrition was regarded as the essential inclusion criterion. We included only patients who had been diagnosed in our center or referred to us before the start of the GFD to evaluate better adherence to follow-up (patients diagnosed in other centers then coming for a follow-up evaluation years after the first diagnosis were not considered). Patients with incomplete medical records or unconfirmed diagnoses were excluded from this study. Finally, patients with complicated CD (i.e., refractory CD, ulcerative jejunoileitis, hyposplenism, lymphoma, and small-bowel carcinoma) were also excluded due to the obvious repercussions on follow-up adherence (Figure 1).

All remaining patients were considered eligible for the evaluation of the following data: (1) age at diagnosis of CD (categorized as <40 years, 40–59 years, and ≥60 years—as proposed by Biagi and colleagues) [17], (2) sex, (3) family history of CD, (4) reason leading to diagnosis of CD (symptom-detected vs. screening-detected cases), (5) clinical presentation (classical vs. nonclassical), (6) iron deficiency, and (7) osteopenia/osteoporosis at diagnosis. In particular, we verified whether the rate of early drop-out from follow-up procedures (defined as follow-up duration <6 months) was different across the categories mentioned above.

In addition, only for patients who performed at least two evaluations (one of which at least six months after the beginning of the GFD), we investigated the correlations between the total length of follow-up and the following additional variables: GFD adherence, status of non-responder CD defined according to the Oslo classification [1], and the presence of metabolic alterations induced by the GFD (body weight increase >10 kg, hypercholesterolemia not present during the gluten-containing diet, development of overt metabolic syndrome).

### 2.3. Evaluations

The clinical evaluations were scheduled according to the Italian Guidelines (a first follow-up visit 6 months after the beginning of the GFD, then every 18–24 months) [15]. Patients were systematically educated about the GFD at diagnosis and at each follow-up visit by the physicians of the clinics. In particular, we provided information about which food contains gluten, which might be contaminated by gluten, and which were safe and permitted. This educational process was similar for younger and older patients. All physicians had a minimum of 5 years of expertise in the management of CD and GFD.

Data about medical history, physical examination, and laboratory tests were gathered on the occasion of each visit.

Inadvertent or voluntary gluten ingestions, modifications of gluten-related symptoms (including diarrhea, bloating, dyspepsia, skin rash, myalgias, oral aphthous lesions), the appearance of new symptoms, and concurrent therapies were specifically investigated during each visit and recorded accordingly.

Physical examination included the evaluation of vital parameters and an examination of the neck, thorax, and abdomen.

Laboratory tests included: blood cells count, serum ferritin, calcium, glucose, total cholesterol, alanine and aspartate aminotransferases, thyroid-stimulating hormone, anti-tissue transglutaminase immunoglobulin A antibodies (tTGA), and deamidated gliadin peptides immunoglobulin G antibodies (DGPG). Additional tests were performed at diagnosis of CD (in particular other organ- and non-organ-specific antibodies) or on a clinical basis in selected cases (for instance, second-level metabolic or osteometabolic test). Iron-deficiency anemia was defined as hemoglobin below the normal range in the setting of hypoferritinemia and elevated transferrin.

A dual-energy X-ray absorptiometry (DEXA) was prescribed at diagnosis and evaluated during the first follow-up evaluation. Osteopenia was defined as a T-score between −1.0 and −2.5 and osteoporosis as a T-score < −2.5.

### 2.4. Compliance with the GFD

Patients were considered to be adherent to the GFD if they satisfied the following criteria: (1) absence of self-reported accidental or intentional gluten ingestions; (2) remission of all CD-related symptoms [13]; (3) normal tTGA levels [24]; (4) Biagi score > 2 [25].

### 2.5. Compliance with Follow-Up Procedures and Drop-Out Definition

We distinguished early drop-out (patients who never performed a follow-up evaluation with a total absence of follow-up data) and late drop-out (patients who had performed at least a follow-up evaluation). This distinction was necessary for correct data analysis since early drop-out cases lacked follow-up data.

Patients were considered late drop-out if they skipped at least two consecutive follow-up examinations. The General Register Office was consulted for all drop-out cases to ensure that drop-out was not due to patient death. In the case of death, the length of follow-up was censored at the time of the last evaluation. In all other cases, time to drop-out was defined as the interval from the first to the latest evaluation. Patients who had performed a follow-up visit within 24 months prior to the data lock were censored at the time of the latest evaluation.

### 2.6. Ethics

This study was approved by the Institutional Review Board of the Bologna Authority S.Orsola-Malpighi Hospital (Protocol 243/2013/O/OssN) and performed according to the Declaration of Helsinki guidelines. Informed consent was obtained according to Institutional Review Board instructions.

### 2.7. Statistical Analysis

Distribution of continuous variables was assessed with a Shapiro–Wilk test, which showed non-normal distributions. Consequently, continuous variables were expressed as median and interquartile range. Categorical variables were expressed as frequencies. Group comparisons were subsequently performed using the Mann–Whitney test for continuous variables and the two-tailed Fisher’s test for categorical variables. The log-rank test and Cox proportional hazard models were used to evaluate the relationship between the persistence to follow-up procedures and other clinical variables of interest. To avoid time-dependent biases (for instance, a false correlation between metabolic alterations and persistence to the procedures due to the fact that longer follow-up favors the detection of metabolic alterations overtime), all events occurring during follow-up were considered time-dependent covariates.

Variables for which the association in the univariate analysis was *p* < 0.10 were entered into the multivariate models. A value of *p* < 0.05 was considered the cut-off for statistical significance. All statistical analyses were performed with SPSS version 23.0 (SPSS Inc., Chicago, IL, USA).

## 3. Results

### 3.1. Study Population: Baseline Characteristics

A total of 578 patients were included in this study. Most of them were females, younger than 40 years at diagnosis (range: 18–81 years) (Table 1). Patients were observed for a median time of 5.0 years (interquartile range: 2.4–7.9). The total observation period was 3078 patient-years.

### 3.2. Risk of Early Drop-Out

A total of 40 patients (6.9%) were immediately lost to the follow-up. None of the baseline characteristics were predictive of an early drop-out. In particular, the rate of early drop-out was similar in females and males (6.8 vs. 7.2%, *p* = 0.844) and across all age groups (7.3%, 5.5%, and 8.8% in patients aged less than 40 years, 40–60 years and more than 60 years at diagnosis, *p* = 0.503). Additionally, we found no differences between patients with classical vs. nonclassical presentation (10.3% vs. 6.2%, *p* = 0.140), symptomatic vs. asymptomatic diagnosis (7.3% vs. 5.2%, *p* = 0.540), with or without iron-deficiency anemia (8.4% vs. 5.6%, *p* = 0.250).

### 3.3. Follow-Up

Overall, 538 patients (93.1%) had a minimum 6-month follow-up. We analyzed the incidence of clinical problems detected in the follow-up in this population (Table 2).

The vast majority of patients were correctly compliant with the GFD (*n* = 468, 87.0%). The prevalence of irritable bowel syndrome (IBS)-like and gastroesophageal reflux syndrome (GERD)-like symptoms was 13.9 and 4.5%, respectively. Metabolic alterations appeared in 20.3% of patients. Osteoporosis was diagnosed in 44.2% of patients. The onset of these problems was usually within the first two years of the GFD (Figure 2), with a median time to onset of 1.9 (IQR: 0.7–3.3), 2.6 (IQR: 0.9–6.9), 2.0 (IQR: 0.8–5.8), and 1.0 (IQR: 1.0–4.0) for IBS-like symptoms, GERD-like manifestations, metabolic alterations, and osteoporosis, respectively.

### 3.4. Risk of Late Drop-Out

A total of 272 patients (50.6%) eventually dropped from follow-up during the observation period. The median time to drop-out was 7.4 years (95% confidence interval: 6.8–8.0).

Age at diagnosis, classical presentation, and metabolic bone disease were associated with time to drop out in the univariate analysis. Multivariable Cox analysis confirmed that classical presentation and age at diagnosis were independently associated with a reduced likelihood of dropping out (Table 3).

After stratifying for the mentioned factors, the median length of follow-up was 6.6 (95% CI 5.8–7.5), 12.8 (6.8–18.8), and 7.8 (CI not evaluable) years in patients aged less than 40, 40–60, and more than 60 years at diagnosis, respectively. Additionally, time to drop-out was 14.0 vs. 7.2 years in patients with classical and nonclassical presentation, respectively (Figure 3).

## 4. Discussion

We reported adherence to follow-up procedures for CD in a tertiary referral center.

The main finding of our study is that high-risk groups are more compliant with follow-up procedures compared with younger patients without a classical presentation.

More importantly, our results showed that the recommendations provided by the guidelines, even if respected by the physicians (in terms of patient education and scheduling of the next follow-up visit) might not find a full application due to patients dropping out of the evaluations.

Previous to our study, Pekki et al. [26] explored the role of follow-up procedures in a Finnish population. Through newspaper advertisements and CD societies’ aid, the authors gathered 677 patients who had received their diagnosis years before (median time to diagnosis: ten years).

According to local policies, patients were referred to the primary care centers after diagnosis. Among them, 84 (12%) had not received any follow-up, 465 (69%) were followed up for less than two years, and 99 (15%) had received long-term follow-up. This pattern was in stark contrast with the local guidelines, which recommended long-term follow-up for all patients [27].

Interestingly, 98% of patients with regular long-term follow-up wished for it also in the future, and this was also seen in over 80% of patients not under follow-up. Patients had no preference about who should be in charge of follow-up [26]. The authors found follow-up was not predicted by gender, age at diagnosis, clinical presentation, or GFD adherence. However, they underlined that the design of their study could be subject to a relevant selection bias.

Our study differs from Pekki et al. [26] in design, a difference mainly dictated by the different organization of follow-up procedures in Finland and Italy (with CD patients being followed up in primary and secondary centers, respectively). Our study did aim to determine which follow-up strategy is the best.

Instead, our data confirmed that most events related to GFD management occur in the first two years (including IBS-like and GERD-like symptoms, the first diagnosis of osteoporosis, and appearance of metabolic alterations). This figure, paired with the knowledge that most cases of complicated CD are diagnosed within the first two years for diagnosis of CD [17], suggests that follow-up shorter than two years might not be in the best interest of patients, as they would be deprived of a medical support at a time in which problems related either to the CD or GFD are more likely to appear.

Despite providing data gathered in a large cohort of patients followed up for an extended timeline, our study has some limitations that need to be discussed. First, our data came from the perspective of a third-level center and did not include patients diagnosed and followed up on different levels of healthcare. As mentioned earlier, this hospital-center approach is the standard in Italy, but not in other countries where CD patients are mainly transferred to the primary care setting. While this situation limits the generalisability of our study, this problem is intrinsic in most studies, as there are no specific recommendations about which setting is the best to perform an optimal follow-up. Additionally, studies involving different levels of healthcare are tough to perform without using surveys (a choice that would expose to other and more relevant biases). Second, even if most second-level Italian centers strictly adhere to the guidelines for the management of CD, our data are monocentric in nature.

While no elements suggest that the adherence of patients or the education provided by the physicians are different across our country, caution should be adopted in interpreting our results.

Third, the study design did not allow the investigators to assess the cause of the drop-out, with the exception of death. As a matter of fact, the General Register Office could be consulted, but the patients could not be contacted directly without breaching the local laws to protect the privacy data.

While death can be a cause of drop-out more represented in the elderly patients, other causes of drop-out should be equally distributed across the study groups. Thus, this limitation is unlikely to have affected our findings. Similarly, the relatively small number of censored observations among the most recently enrolled patients is unlikely to have significantly affected our analysis of late drop-out, as a minimum theoretical three-year observation was required according to our study protocol. Finally, it is not possible to state whether our patients continued to be followed up in the primary care setting. Previous data, however, showed that the rate of patients receiving follow-up in the primary care setting could be abysmal. Besides the mentioned study of Pekki and colleagues (which found a rate of proper follow-up of 15% in Finland) [26], Bebb et al. [28] reported only 6% of patients diagnosed in the secondary centers and later transferred to primary care received follow-up in the United Kingdom.

Despite these limitations, our study also has elements of strength. In comparison with previous studies, which used questionnaires to assess patient GFD adherence, our study avoided some important selection biases (for example study information reaching different age groups and socioeconomic classes unequally, selection of attention-seeking patients). Most importantly, serological follow-up was available for all patients. Finally, the use of time-dependent variables contributed to avoiding spurious correlations (such as patients with metabolic alterations or IBS-like manifestation falsely appearing to have longer follow-up) and correctly assessing causality.

In conclusion, we reported for the first time that adherence to follow-up procedures is heterogeneous (with patients at increased risk of complications being less likely to drop out from follow-up visits), marking a difference from the scenario defined by the guidelines, which envisions a similar follow-up for all patients.

These data suggest that the academic hypothesis of adopting different follow-up lengths based on the risk of complicated CD is a fact in everyday clinical practice and is driven by the adherence of different groups of patients, rather than by medical prescriptions. While the current guidelines suggestion of visits at 0, 6, and 18–24 months envision a good model for education and evaluation of results, their actual application remains an open problem, with healthcare professionals needing to continuously motivate patients toward follow-up procedures (and not merely prescribing controls). Future studies of the cost-effectiveness of follow-up procedures in primary or secondary prevention of complicated CD are needed to understand the most appropriate scenario better and allocate National Health Service funds.

## Figures and Tables

**Figure 1 nutrients-14-01223-f001:**
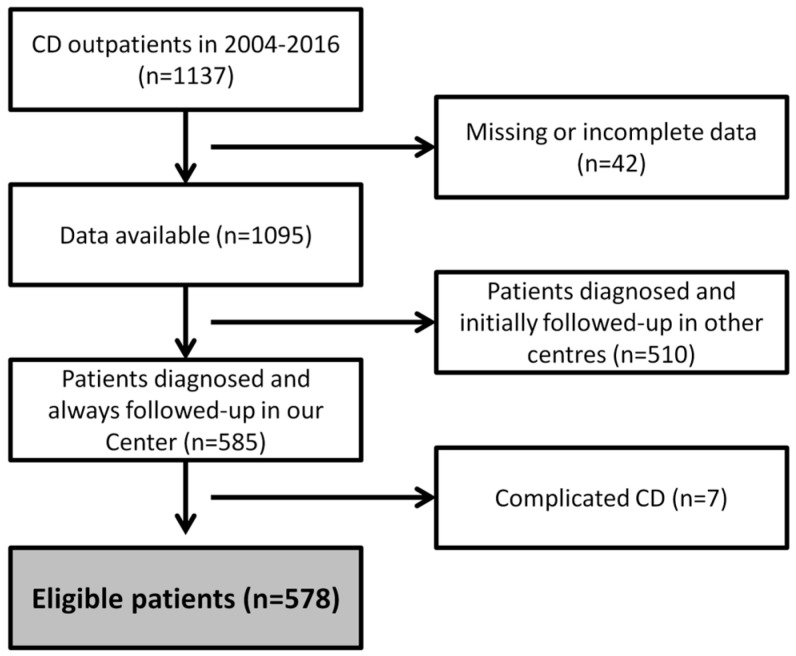
Patient disposition.

**Figure 2 nutrients-14-01223-f002:**
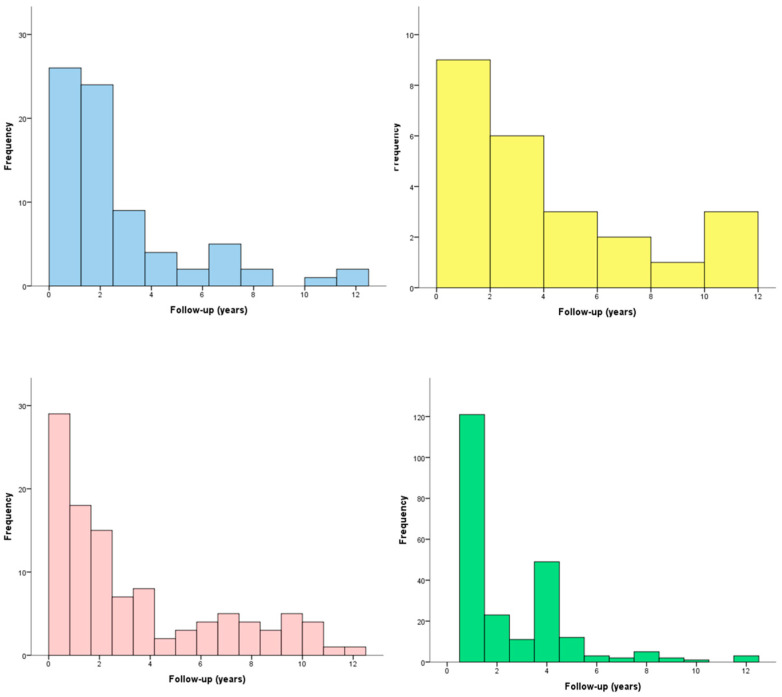
Time to occurrence of various events during follow-up of celiac patients, including irritable bowel syndrome-like symptoms (blue), gastroesophageal reflux disease-like manifestations (yellow), metabolic alterations (red), and osteoporosis (green).

**Figure 3 nutrients-14-01223-f003:**
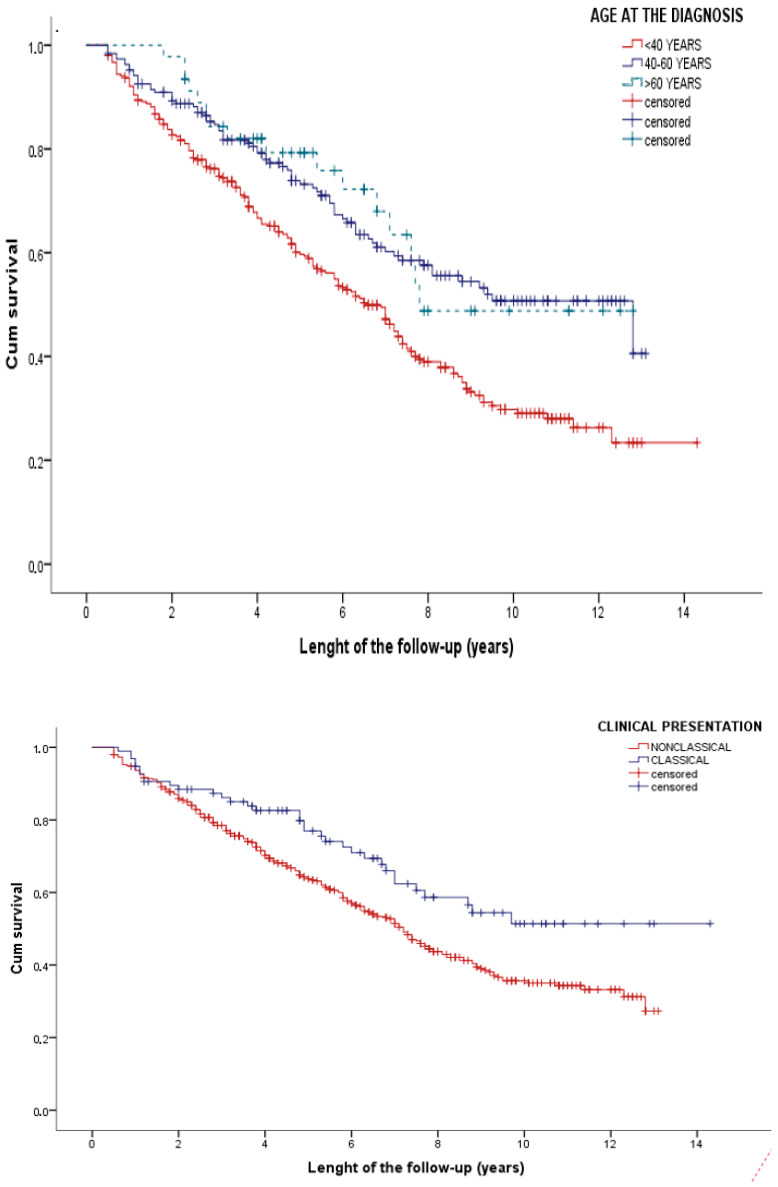
Time to drop-out stratified according to (**above**) age at diagnosis; (**below**) clinical presentation of celiac disease.

**Table 1 nutrients-14-01223-t001:** Characteristics of patients with celiac disease (CD) at diagnosis (*n* = 578). Variables are expressed as frequencies (percentage).

Variable	
Female sex	453 (78.4)
Age at diagnosis	
40 years	327 (56.6)
40–59 years	200 (34.6)
≥60 years	51 (8.8)
Family history of CD	109 (18.9)
Symptomatic CD	471 (81.5)
Classical presentation	107 (18.5)
Iron-deficiency anemia	275 (47.6)

**Table 2 nutrients-14-01223-t002:** Events occurring during follow-up of CD patients who did not drop out in the first 6 months (*n* = 538). Variables are expressed as frequencies (percentage).

Variable	
GFD adherence	468 (87.0)
IBS-like symptoms (total)	80 (14.9)
* IBS-mixed-like symptoms	44 (8.2)
* IBS-diarrhea-like symptoms	9 (1.7)
* IBS-constipation-like symptoms	27 (5.0)
GERD-like symptoms	24 (4.5)
Metabolic alterations (at least one)	109 (20.3)
* Weight gain > 10 kg	55 (10.2)
* Hypercholesterolemia	59 (11.0)
* Metabolic syndrome	25 (4.6)
Osteoporosis	238 (44.2)

GFD: gluten-free diet IBS: irritable bowel syndrome; GERD: gastroesophageal reflux syndrome. * Analyzed as time-dependent variables.

**Table 3 nutrients-14-01223-t003:** Predictors of risk of late drop-out according to the multivariable Cox model with time-dependent analysis.

Univariable Analyses				Multivariable Analyses		
Hazard Ratio	95% CI	*p*		Hazard Ratio	95% CI	*p*
0.994	0.738–1.339	0.880	Female sex			
			Age at diagnosis			
*-*	Reference	*-*	<40 years	*-*	Reference	*-*
0.583	0.444–0.765	<0.001	40–59 years	0.600	0.455–0.790	<0.001
0.556	0.333–0.927	0.024	≥60 years	0.598	0.357–0.997	0.048
0.900	0.659–1.230	0.510	Family history of CD			
1.235	0.906–1.683	0.181	Symptomatic CD			
0.648	0.456–0.920	0.015	Classical presentation	0.641	0.446–0.920	0.016
1.019	0.803–1.292	0.880	Iron deficiency			
0.761	0.597–0.970	0.027	Osteoporosis *	0.846	0.659–1.086	0.190
1.317	0.944–1.836	0.105	GFD compliance *			
1.356	0.934–1.968	0.190	IBS-like symptoms *			
0.966	0.397–2.349	0.939	GERD-like symptoms *			
0.865	0.579–1.291	0.477	Metabolic alterations *			

* Analyzed as time-dependent variables.

## Data Availability

The data presented in this study are available on request from the corresponding author. The data are not publicly available due to privacy reasons.

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
