# Peer review of "Risk of Drop-Out from Follow-Up Evaluations for Celiac Disease: Is It Similar for All Patients?"

_nutrients, 2022, doi:10.3390/nu14061223_

Round 1

Reviewer 1 Report

This manuscript presents important and relevant aspects to analysis of the management of celiac disease. However, there are some comments that require consideration.

Title:

"Adherence to the follow-up procedures of celiac disease: do all patients behave in the same way?".

The choice and the definition of “behave” is unclear. Is the intention to refer to how the patients carry out the follow up in terms of adhering to coming to the follow up? Showing up or dropping out of follow up sessions? Or possibly patterns of behavior? This is unclear, as to my understanding, records were retrospectively analyzed, yet the patients’ behavior was not assessed in the study. What measures were used to assess their behavior? Would maybe using the term “characteristics” or “patterns” be more suitable?

Introduction

Lines 37-38: This is the only reference to social impact of CD. However, it seems out of place in the sequence of the introduction. As mentioned re the title, the social impact could be extremely relevant to behavior.

Lines 40-41, lines 60-64, lines 75-77: Here and throughout the article does the follow-up protocols refer to what is expected from the patient or what is expected from the physician/medical staff/health providers? Is responsibility of follow up or drop out on the patient? Are these protocols provided to the patients? Do patients have protocols (Line 56-57). Or are these protocols intended for the physician/medical staff/health providers, hoping that the patients will abide to the instructions they are given? What is the role of the professionals in the adhering to the follow-up procedures? This is needs to be focused and explained to improve clarity of the study, results and conclusions.

Lines 45-55: The authors discuss “complicated CD”, the frequency and need for stricter follow-up, however a definition of complicated CD and its characteristics is lacking. Also in the methods section (lines 93-94).

Lines 65-67: The choice of the word “peculiarities” seems peculiar. Also. “until a few years ago” – how does this fit in with the timeline of the study data? The use of the terms one-level referral centers, multidisciplinary hospital teams, and in the discussion (lines 224, 267, & 275) should be clarified for non-Italian readers.

Materials and Methods

Line 100: “familiarity for CD” – do you mean “familiarity with CD”, “knowledge about CD”? How was the this evaluated?

Line 104: Typo – “foll ow-up” should be follow-up.

Lines 106 – 112: The use of the word “Instead” is unclear and consequently the rest of the sentence. Should it possibly be “In addition” or “Therefore”? Is this analysis of correlations related to the inclusion criteria?

Line 135: “1) absence of accidental or intentional gluten ingestions” – how was this assessed?

Line 152: The data used were from records (2004-2017). Did patients sign informed consent during those years? Where all patients approached and asked to sign?

Results

Line 182 and in Table 1: What was the definition of classical vs non-classical presentation?

Line 191: “compliant to the GFD”? How was this assessed? Based on what criteria?

Discussion

Lines 225-233, 241-254: I suggest considering first incorporating these sections to the introduction of the article and not the discussion. If presented in the introduction, the results of course can be discussed while referencing to this literature.

 Lines 237-239: What exactly is the physician’s perspective? And what is the patient’s behavior? What is the meaning of this statement?

Lines 259-264: Does this refer to the patients’ behavior or to the physician’s conduct in carrying out follow-up?

Lines 273-274: “tough to perform without using questionnaires” – yet this study claims to examine the patients behavior, without asking them questions about there behavior? This is unclear.

Line 278: country should not be with a capital “c”.  

Line 289: Is this referring to Italy?

Author Response

Point-byPoint reponse to the reviewer's comments (Please see also the attachment)

Reviewer #1

Title:

"Adherence to the follow-up procedures of celiac disease: do all patients behave in the same way?".

The choice and the definition of “behave” is unclear. Is the intention to refer to how the patients carry out the follow up in terms of adhering to coming to the follow up? Showing up or dropping out of follow up sessions? Or possibly patterns of behavior? This is unclear, as to my understanding, records were retrospectively analyzed, yet the patients’ behavior was not assessed in the study. What measures were used to assess their behavior? Would maybe using the term “characteristics” or “patterns” be more suitable?

Author response: We agree with the reviewer that the terms “behave”, and “behavior” might be inappropriate in this setting. It was our intention to refer to how the patients carry out the follow up in terms of adhering to coming to the follow up (showing up or dropping out of follow up sessions). We changed the title as follows: “Risk of drop-out from the follow-up evaluations for celiac disease: is it similar for all patients?”

 Introduction

Lines 37-38: This is the only reference to social impact of CD. However, it seems out of place in the sequence of the introduction. As mentioned re the title, the social impact could be extremely relevant to behavior.

Author response: We agree with the referee comment on the relevance of social-economic impact of GFD. We now report an additional reference. However, the Italian clinical setting is characterized by some specificities. We also report results of a large Italian study demonstrating that economic resources allocated monthly to patients are quite adequate, thus suggesting that they do not impact compliance with GFD (reference 23).

Lines 40-41, lines 60-64, lines 75-77: Here and throughout the article does the follow-up protocols refer to what is expected from the patient or what is expected from the physician/medical staff/health providers? Is responsibility of follow up or drop out on the patient? Are these protocols provided to the patients? Do patients have protocols (Line 56-57). Or are these protocols intended for the physician/medical staff/health providers, hoping that the patients will abide to the instructions they are given? What is the role of the professionals in the adhering to the follow-up procedures? This is needs to be focused and explained to improve clarity of the study, results and conclusions.

Author response: We agree with the reviewer about the opportunity of improved clarity.  The protocols are intended for the physician/medical staff/health providers, hoping that the patients will abide to the instructions they are given. We modified the unclear sentences as requested by the reviewer:

“As a consequence of these possible difficulties, all guidelines for CD provide physicians and other healthcare professionals with recommendations about the follow-up for CD patients. (10–14)

“The study of the adherence of patients to the recommendations provided by the healthcare professionals can be complex in Countries in which CD patients are diagnosed and followed up by a wide range of physicians”

“In such a setting, we aimed to identify the factors which can influence and affect the short and long-term adherence of the patients to the follow-up evaluations prescribed by their physicians according to the current guidelines”. 

Lines 45-55: The authors discuss “complicated CD”, the frequency and need for stricter follow-up, however a definition of complicated CD and its characteristics is lacking. Also, in the methods section (lines 93-94).

 Author response: Thanks for this suggestion. We provided a full description of the possible forms of complicated CD, both in the introduction and methods: “including refractory CD, ulcerative jejunoileitis, hyposplenism, lymphoma, and small-bowel carcinoma”.

Lines 65-67: The choice of the word “peculiarities” seems peculiar. Also. “until a few years ago” – how does this fit in with the timeline of the study data? The use of the terms one-level referral centers, multidisciplinary hospital teams, and in the discussion (lines 224, 267, & 275) should be clarified for non-Italian readers.

Author response: Thanks for these suggestions. We modified our sentence as follows: “The Italian clinical setting, however, is characterized by some specificities. Until a few years ago, the diagnosis of CD had to be confirmed by physicians working in regional referral centers to grant the patients the possibility of obtaining a refund for gluten-free products approximately worth 100 euros monthly. At the same time, most of these centers had a policy to follow-up the patients indefinitely and in collaboration with their general practitioners, rather than merely referring them back after a fixed time. In the last few years, hospital teams consisting of gastroenterologists, pathologists, nutritionists, and rheumatologists have been created to improve the reliability of the diagnostic process…”

 Materials and Methods

Line 100: “familiarity for CD” – do you mean “familiarity with CD”, “knowledge about CD”? How was the this evaluated?

 Author response: We meant “family history of CD”. We have modified the sentence.

Line 104: Typo – “foll ow-up” should be follow-up.

 Author response: Thanks. Modified as suggested.

Lines 106 – 112: The use of the word “Instead” is unclear and consequently the rest of the sentence. Should it possibly be “In addition” or “Therefore”? Is this analysis of correlations related to the inclusion criteria?

Author response: Thanks for pointing out our error. We intended “In addition”. The sentence has been corrected.

Line 135: “1) absence of accidental or intentional gluten ingestions” – how was this assessed?

Author response: Point 1 intended ingestions referred by the patients. Subsequent points provided additional criteria to increase the sensitivity. For instance, a patient who referred gluten ingestions was considered as non-compliant (either because he/she did it as a voluntary transgression or experienced and recognized accidental contaminations). Also, a patient denying ingestions but with recurring CD symptoms and positive serology was equally considered as non-compliant (he/she probably ate gluten-containing foods not knowing sufficient information about their actual nutritional composition). For clarity, we modified point 1 as follows: “absence of referred accidental or intentional gluten ingestions”.

Line 152: The data used were from records (2004-2017). Did patients sign informed consent during those years? Where all patients approached and asked to sign?

Author response: All patients, after being informed, signed the informed consent for the collection of clinical data that are suitable in performing clinical practice studies during those years.

Results

Line 182 and in Table 1: What was the definition of classical vs non-classical presentation?

 Author response: Classical presentation was defined according to the Oslo definitions for CD. We now added this information in the Introduction: “…classical presentation (i.e. with signs and symptoms of malabsorption according to the Oslo definitions…”

Line 191: “compliant to the GFD”? How was this assessed? Based on what criteria?

Author response: The criteria for defining patients as compliant to the GFD have been reported in Paragraph 2.4. No single criterion is sufficiently sensitive to detect gluten consumption, while all of the criteria we used are validated in literature: “Patients were considered to be adherent to the GFD if they satisfied all of the following criteria: 1) absence of referred accidental or intentional gluten ingestions; 2) remission of all CD-related symptoms (12);  3) normal tTGA levels (22); 4) Biagi score>2 (23)”

Discussion

Lines 225-233, 241-254: I suggest considering first incorporating these sections to the introduction of the article and not the discussion. If presented in the introduction, the results of course can be discussed while referencing to this literature.

Author response: Thanks for this suggestion. We moved these informations in the “Introduction”.

Lines 237-239: What exactly is the physician’s perspective? And what is the patient’s behavior? What is the meaning of this statement?

Author response: We agree that our sentence was confusing, especially when dealing with the term “behavior”. We modified our sentence as follows: “More importantly, our results showed that the recommendations provided by the guidelines, even if respected by the physicians (in terms of patients education and scheduling of the next follow-up visit) might not find a full application due to patients dropping out of the evaluations.”

Lines 259-264: Does this refer to the patients’ behavior or to the physician’s conduct in carrying out follow-up?

Author response: Once again, we thank the reviewer for pointing out a potentially unclear sentence. We referred to both patients and physicians. Also due to another reviewer objecting on the same sentence, we rephrased it as follows: “a follow-up shorter than two years might not be in the best interest of the patients as they would be deprived of a medical support at a time in which problems related either to the CD or GFD are more likely to appear.”

Lines 273-274: “tough to perform without using questionnaires” – yet this study claims to examine the patient behavior, without asking them questions about their behavior? This is unclear.

Author response: Thanks for pointing out an unintentional mistake. The sentence has now been corrected: “studies involving different levels of healthcare are tough to perform without using surveys (a choice that would expose to other and more relevant biases).”

Line 278: country should not be with a capital “c”.  

Author response: Thanks for the suggestion. The word has been corrected.

Line 289: Is this referring to Italy?

Author response: The studies by Bebb and Pekki were performed in other countries. We now specified this aspect: “Besides the mentioned study of Pekki and colleagues (which found a rate of proper follow-up of 15% in Finland)(24), Bebb et al. (26) reported only 6% of the patients diagnosed in the secondary centers and later transferred to primary care received follow-up in the United Kingdom.

Reviewer #2

Title: Adherence to the follow-up procedures of celiac disease: do all patients behave in the same way?

The study focuses on the problem of adherence to the follow-up recommendations among patients suffering from celiac disease. The article is interesting and covers important aspects of the management of patients requiring gluten-free diet. However, the authors analyze only selected parameters of the follow-up without a detailed explanation of the methodology used for data collection.

Please address the following issues:

The title of the article is not appropriate for its content, as the authors do not define or evaluate the behavior of people with CD. The answer to the authors’ question: do all patients behave in the same way (?) is no, regardless the diseases they suffer from. Please consider changing the title.

Author response: We agree with this reviewer that the previous title was potentially misleading. As a matter of fact, our aim was to verify how many patients dropped out from the follow-up visits and whether specific risk factors were linked to the drop-out. In particular, we intended to verify whether the risk factors for complicated CD were also factors increasing/decreasing the risk of drop-out. Consequently, we modified the title as follows: “Risk of drop-out from the follow-up evaluations for celiac dis-ease: is it similar for all patients?”

 The results provided in the abstract are related only to the drop-out of patients, and not the adherence to follow-up procedures

 Author response: As mentioned in the previous point, adherence to the follow-up procedure was measured as persistence to the follow-up evaluations. Consequently, a drop-out was considered as a lack of adherence. Sorry if this point was not made clear in the original version of our manuscript. 

Please provide information on the criteria for defining the patients at increased risk of complication/complicated CD (line 17)? 

Author response: Thanks for pointing out the relative lack of information in this regard in the original version of our manuscript. We added the following lines, detailing the results from a keystone paper regarding the risk complicated CD

“In particular, Biagi et al (17) reported that patients older than 60 years at the diagnosis of CD had a 18-fold risk of complication compared with patients diagnosed at 18-40 years and a 9 times higher risk than patients diagnosed at 40-60 years. Classical presentation also increased the risk of complications by 7 times compared to non-classical presentation (17)”

 Please give more detail about the criteria of adherence to the follow-up and drop-out.

Author response: The main clinical guidelines for CD describe the correct follow-up procedure as follows: “Currently, the suggested protocols call for a first examination six months after the beginning of the GFD and every 18-24 thereafter”. The guidelines do not provide universal definitions for “drop-out” as the setting for the follow-up may vary a lot from one National Health System from another.  For the purpose of our study and according to the organization of of our National Health System (which arranges for gastroenterology outpatient evaluations), we considered as drop-outs patients who had no follow-up visits at all after the diagnosis or patients who skipped at least two consecutive follow-up examinations. We reported this information in the Introduction and in Paragraph 2.5 (whose title has been modified as “Compliance to the follow-up procedures and drop-out definition”).

 The groups distinguished by age at diagnosis are not equal. Did the authors consider another division into groups (e.g., into quartiles of age)? What was the youngest age of patient inclusion?

 Author response: The reviewer is absolutely right in this objection, as we forgot to clearly mention the definition of “patients at high-risk of recurrence” (see previous point). The division of the age groups was based on the evidence emerged from the paper by Biagi et al, which stratified the risk of complications as different according to following age at presentation: 18-39; 40-59, 60 and higher. Since our main aim was to verify the adherence in various groups with different risk of complications, we adopted the same division into subgroups. Dividing according to quartiles would have provided groups with a similar number of patients but threshold of ages which were never investigated nor validated (thus preventing any reliable comparison with previous evidence). We also added more information about the youngest and eldest age of patients’ inclusion in the text “(range 18-81 years)”. 

 Please explain how long was the period of the observation and the total length of follow-up. Patient enrollment was between 2004 and 2016 and the database was locked in 2020. Authors stated (lines 82-84) the final cut-off date was chosen to allow a theoretical minimum three-year follow-up even for the most recently diagnosed patients. However, for some patients, the period of follow-up appears to be potentially much longer. In my opinion these are so-called censored observations, which should be described as limitations of the study. The authors describe the use of time-dependent variable as a strong point (lines 297-299).

Author response: Indeed, the adherence to the follow-up procedures was evaluated according to a survival analysis in which a missed visit acted as an event. Similarly to most of the survival analysis, patients started observation at different times and had potentially different length of follow-up (even in randomized clinical trials, patients are enrolled at different times and the database is locked at a fixed time, generating different lights of observation)- The reviewer is right in considering that part of the most recent patients had censored observations, but this is quite normal in this kind of analyses. Still, we agree that the median time of observation is an important information which was missing in our manuscript. We added the following specification:” Patients were observed for a median time of 5.0 years (interquartile range 2.4-7.9). The total observation period was 3078 patient-years.”

 The problem mentioned above is also important for the assessment of the risk of late drop-out (line 302) and the analysis presented in table 3.

 Author response: We added the following text to the limitations paragraph. “Similarly, the relatively small number of censored observations amongst the most recently enrolled patients is unlikely to have signicantly affected our analysis of late drop-out as a minimum theoretical three-year observation was required according our study protocol”

 Please explain the criteria of iron deficiency anemia and osteopenia/osteoporosis diagnosis. Were these diseases age or sex-dependent?

Author response: We added the following specifications:

“Iron deficiency anaemia was defined as hemoglobin below the normal range in the setting of hypoferritinaemia and elevated transferring”

“Osteopenia was defined as a T-score between -1.0 and -2.5 and osteoporosis as a T-score < -2.5.”

Both conditions have a different prevalence according to age and sex, but the multivariable analysis did not find VIF values suggestive of relevant collinearity between age and sex on one hand and osteopenia/osteoporosis and iron deficiency anemia on the oth.er hand. As a consequence, possible differences were addressed by the multivariable analysis itself

 Similarly, criteria of diagnosis for all diseases/conditions listed in table 2 should be clarified. Table 2 provides information on concomitant disorders or diseases rather than events.

Author response: Actually, Table 2 reports symptoms (not concomitant disorders) which appeared during the follow-up. This the case for IBS- and GERD-like symptoms, which are a most common occurrence after the start of a gluten-free diet (please see reference 10). Also, hypercholesterolemia and metabolic syndrome are not concurrent conditions at the start of the GFD, but rather novel conditions appearing favoured by the GFD itself (reference 6-9).

Body weight increased > 10 kg was one of the criteria of metabolic alterations induced by GFD. However, higher body mass is associated with positive energy balance and general overconsumption, not necessarily related to GFD. Increased body weight could be also related to age of patients.

Author response: We are aware that the existing criteria defining metabolic alterations appearing after the start of the GFD has some intrinsic limitations. Still, any alteration which was not present at the start of the GFD and which appeared thereafter is usually considered as GFD-related, provided that it appears within a reasonable amount of time. In most studies, metabolic alterations appear within the first 12-24 months after the start of the GFD in the majority of patients. Our findings largely confirm these data (Figure 3). As such, we shared the common interpretation of metabolic alterations induced by to a positive energy balance due to an unfavorable nutritional contents of the gluten-free foods (reference 6-9).

 Please provide explanation how was the compliance to GFD measured. Please provide the information about the methods of the analysis of the quality of patients’ diet. In the methodology there is only a statement (line 135): absence of accidental or intentional gluten ingestions, and in the result section there is only a short information (line 191) The vast majority of the patients were correctly compliant to the GFD (n=468, 87.0%). Please provide explanation who, when and using what methodology assessed the compliance to the GFD.

 Author response: Actually, four different criteria were adopted to analyses adherence, hot merely absence of accidental or intentional gluten ingestions, but also remission of all CD-related symptoms; 3) normal tTGA levels; 4) Biagi score>2 (please refer to paragraph 2.4). All of these parameters are widely accepted criteria to assess compliance to the GFD, as a gold-standard is absent. The assessments were performed during the evaluations according to the same methodology reported in the referenced papers.

 To follow GFD patients have to be educated about the principles of the diet. Please explain how patients were educated and who educated the patients. Was education similar for younger and older patients?  

Author response: We added the following specification: “Patients were systematically educated about GFD at the diagnosis and at each follow-up visit by the physicians of the clinics. In particular, we provided information about which foods contains gluten, which might be contaminated by gluten, and which were safe and permitted. This educational process was similar for younger and older patients. All of the physicians had a minimum 5-year expertise in the management of CD and GFD.”

 Please explain prom the patient point of view your statement (lines 263-264).. that a follow-up shorter than two years might not be in the best interest of the patients.

Author response: We added the following specification: “a follow-up shorter than two years might not be in the best interest of the patients as they would be deprived of a medical support at a time in which problems related either to the CD or GFD are more likely to appear”.

Please explain in more detail the sentence (lines 277-278): While no elements suggest that the behavior of patients and physicians is different across our Country, caution should be adopted in interpreting our results. How did the authors measure the behavior of the patients?

Author response: We are sorry if we used the term “behavior” in a way which not completely appropriate for this context. We modified our sentence to avoid any possible misleading message: “While no elements suggest that the adherence of patients or the education provided by the physicians are different across our Country, caution should be adopted in interpreting our results”

 Please explain which study findings are related to the conclusion provided in the lines 300-303 and 304-306.

Author response:  We modified the text as follows:

“In conclusion, we reported for the first time that the adherence to the follow-up procedures is heterogeneous (with patients at increased risk of complications being less likely to drop out from the follow-up visits)”

“…adopting different follow-up lengths based on the risk of complicated CD is actually a fact in everyday clinical practice, which is driven by the patients’ adherence rather than by medical prescriptions.”

 Please give your opinion about the model of the follow-up procedures at 0, 1 month and 18-24 months after CD diagnosis and the time for each visit. Is it an optimal model for education and an evaluation of the results? Is always the problem related to patients? Should health care personnel be evaluated for their effectiveness?

Author response: We are grateful to the reviewer for this suggestion. We added the following considerations: “While the current guidelines suggestion of visits at 0, 6, and 18-24 months envision a good model for education and evalaution of results, their actual application remain an open problem, with healthcare professional needing to continously motivate patients toward the follow-up procedures (and not merely prescribing controls)”. 

I suggest reconsidering your conclusions to ensure they are consistent with the purpose and findings of the study.

Author response: Please refer to the responses from the previous two points.

Reviewer 2 Report

Title: Adherence to the follow-up procedures of celiac disease: do all patients behave in the same way?

The study focuses on the problem of adherence to the follow-up recommendations among patients suffering from celiac disease. The article is interesting and covers important aspects of the management of patients requiring gluten-free diet. However, the authors analyze only selected parameters of the follow-up without a detailed explanation of the methodology used for data collection.

Please address the following issues:

  • The title of the article is not appropriate for its content, as the authors do not define or evaluate the behavior of people with CD. The answer to the authors’ question: do all patients behave in the same way (?) is no, regardless the diseases they suffer from. Please consider changing the title.
  • The results provided in the abstract are related only to the drop-out of patients, and not the adherence to follow-up procedures.
  • Please provide information on the criteria for defining the patients at increased risk of complication/complicated CD (line 17)?
  • Please give more detail about the criteria of adherence to the follow-up and drop-out.
  • The groups distinguished by age at diagnosis are not equal. Did the authors consider another division into groups (e.g., into quartiles of age)? What was the youngest age of patient inclusion?
  • Please explain how long was the period of the observation and the total length of follow-up. Patient enrollment was between 2004 and 2016 and the database was locked in 2020. Authors stated (lines 82-84) the final cut-off date was chosen to allow a theoretical minimum three-year follow-up even for the most recently diagnosed patients. However, for some patients, the period of follow-up appears to be potentially much longer. In my opinion these are so-called censored observations, which should be described as limitations of the study. The authors describe the use of time-dependent variable as a strong point (lines 297-299).
  • The problem mentioned above is also important for the assessment of the risk of late drop-out (line 302) and the analysis presented in table 3.
  • Please explain the criteria of iron deficiency anemia and osteopenia/osteoporosis diagnosis. Were these diseases age or sex-dependent?
  • Similarly, criteria of diagnosis for all diseases/conditions listed in table 2 should be clarified. Table 2 provides information on concomitant disorders or diseases rather than events.
  • Body weigh increased > 10 kg was one of the criteria of metabolic alterations induced by GFD. However, higher body mass is associated with positive energy balance and general overconsumption, not necessarily related to GFD. Increased body weight could be also related to age of patients.
  • Please provide explanation how was the compliance to GFD measured. Please provide the information about the methods of the analysis of the quality of patients’ diet. In the methodology there is only a statement (line 135): absence of accidental or intentional gluten ingestions, and in the result section there is only a short information (line 191) The vast majority of the patients were correctly compliant to the GFD (n=468, 87.0%). Please provide explanation who, when and using what methodology assessed the compliance to the GFD.
  • To follow GFD patients have to be educated about the principles of the diet. Please explain how patients were educated and who educated the patients. Was education similar for younger and older patients?  
  • Please explain prom the patient point of view your statement (lines 263-264).. that a follow-up shorter than two years might not be in the best interest of the patients.
  • Please explain in more detail the sentence (lines 277-278): While no elements suggest that the behavior of patients and physicians is different across our Country, caution should be adopted in interpreting our results. How did the authors measure the behavior of the patients?
  • Please explain which study findings are related to the conclusion provided in the lines 300-303 and 304-306.
  • Please give your opinion about the model of the follow-up procedures at 0, 1 month and 18-24 months after CD diagnosis and the time for each visit. Is it an optimal model for education and an evaluation of the results? Is always the problem related to patients? Should health care personnel be evaluated for their effectiveness?
  • I suggest reconsidering your conclusions to ensure they are consistent with the purpose and findings of the study.

In conclusion, the article requires major revisions.

Author Response

Point-to-point reply

Reviewer #2

Title: Adherence to the follow-up procedures of celiac disease: do all patients behave in the same way?

The study focuses on the problem of adherence to the follow-up recommendations among patients suffering from celiac disease. The article is interesting and covers important aspects of the management of patients requiring gluten-free diet. However, the authors analyze only selected parameters of the follow-up without a detailed explanation of the methodology used for data collection.

Please address the following issues:

  • The title of the article is not appropriate for its content, as the authors do not define or evaluate the behavior of people with CD. The answer to the authors’ question: do all patients behave in the same way (?) is no, regardless the diseases they suffer from. Please consider changing the title.

Author response: We agree with this reviewer that the previous title was potentially misleading. As a matter of fact, our aim was to verify how many patients dropped out from the follow-up visits and whether specific risk factors were linked to the drop-out. In particular, we intended to verify whether th risk factors for complicated CD were also factors increasing/decreasing the risk of drop-out. Consequently, we modified the title as follows: “Risk of drop-out from the follow-up evaluations for celiac dis-ease: is it similar for all patients?”

  • The results provided in the abstract are related only to the drop-out of patients, and not the adherence to follow-up procedures

Author response: As mentioned in the previous point, the adherence to the follow-up procedure was measured as persistence to the follow-up evaluations. Consequently, a drop-out was considedered as a lack of adherence. Sorry if this point was not made clear in the original version of our manuscript.

  • Please provide information on the criteria for defining the patients at increased risk of complication/complicated CD (line 17)?

Author response: Thanks for pointing out the relative lack of information in this regard in the original version of our manuscript. We added the following lines, detailing the results from a keystone paper regarding the risk complicated CD

“In particular, Biagi et al (17) reported that patients older than 60 years at the diagnosis of CD had a 18-fold risk of complication compared with patients diagnosed at 18-40 years and a 9 times higher risk than patients diagnosed at 40-60 years. Classical presentation also increased the risk of complications by 7 times compared to non-classical presentation (17)”

  • Please give more detail about the criteria of adherence to the follow-up and drop-out.

Author response: The main clinical guidelines for CD describe the correct follow-up procedure as follows: “Currently, the suggested protocols call for a first examination six months after the beginning of the GFD and every 18-24 thereafter”. The guidelines do not provide universal definitions for “drop-out” as the setting for the follow-up may vary a lot from one National Health System from another.  For the purpose of our study and according to the organization of of our National Health System (which arranges for gastroenterology outpatient evaluations), we considered as drop-outs patients who had no follow-up visits at all after the diagnosis or patients who skipped at least two consecutive follow-up examinations. We reported this information in the Introduction and in Paragraph 2.5 (whose title has been modified as “Compliance to the follow-up procedures and drop-out definition”).

  • The groups distinguished by age at diagnosis are not equal. Did the authors consider another division into groups (e.g., into quartiles of age)? What was the youngest age of patient inclusion?

Author response: The reviewer is absolutely right in this objection, as we forgot to clearly mention the definition of “patients at high-risk of recurrence” (see previous point). The division of the age groups was based on the evidence emerged from the paper by Biagi et al, which stratified the risk of complications as different according to following age at presentation: 18-39; 40-59, 60 and higher. Since our main aim was to verify the adherence in various groups with different risk of complications, we adopted the same division into subgroups. Dividing according to quartiles would have provided groups with a similar number of patients but threshold of ages which were never investigated nor validated (thus preventing any reliable comparison with previous evidence). We also added more information about the youngest and eldest age of patients’ inclusion in the text “(range 18-81 years)”. 

  • Please explain how long was the period of the observation and the total length of follow-up. Patient enrollment was between 2004 and 2016 and the database was locked in 2020. Authors stated (lines 82-84) the final cut-off date was chosen to allow a theoretical minimum three-year follow-up even for the most recently diagnosed patients. However, for some patients, the period of follow-up appears to be potentially much longer. In my opinion these are so-called censored observations, which should be described as limitations of the study. The authors describe the use of time-dependent variable as a strong point (lines 297-299).

  • Author response: Indeed, the adherence to the follow-up procedures was evaluated according to a survival analysis in which a missed visit acted as an event. Similarly to most of the survival analysis, patients started observation at different times and had potentially different length of follow-up (even in randomized clinical trials, patients are enrolled at different times and the database is locked at a fixed time, generating different lights of observation)- The reviewer is right in considering that part of the most recent patients had censored observations, but this is quite normal in this kind of analyses. Still, we agree that the median time of observation is an important information which was missing in our manuscript. We added the following specification:” Patients were observed for a median time of 5.0 years (interquartile range 2.4-7.9). The total observation period was 3078 patient-years.”

  • The problem mentioned above is also important for the assessment of the risk of late drop-out (line 302) and the analysis presented in table 3.

Author response: We added the following text to the limitations paragraph. “Similarly, the relatively small number of censored observations amongst the most recently enrolled patients is unlikely to have signicantly affected our analysis of late drop-out as a minimum theoretical three-year observation was required according our study protocol”

  • Please explain the criteria of iron deficiency anemia and osteopenia/osteoporosis diagnosis. Were these diseases age or sex-dependent?

Author response: We added the following specifications:

“Iron deficiency anaemia was defined as hemoglobin below the normal range in the setting of hypoferritinaemia and elevated transferring”

“Osteopenia was defined as a T-score between -1.0 and -2.5 and osteoporosis as a T-score < -2.5.”

Both conditions have a different prevalence according to age and sex, but the multivariable analysis did not find VIF values suggestive of relevant collinearity between age and sex on one hand and osteopenia/osteoporosis and iron deficiency anemia on the oth.er hand. As a consequence, possible differences were addressed by the multivariable analysis itself

  • Similarly, criteria of diagnosis for all diseases/conditions listed in table 2 should be clarified. Table 2 provides information on concomitant disorders or diseases rather than events.

Author response: Actually, Table 2 reports symptoms (not concomitant disorders) which appeared during the follow-up. This the case for IBS- and GERD-like symptoms, which are a most common occurrence after the start of a gluten-free diet (please see reference 10). Also, hypercholesterolemia and metabolic syndrome are not concurrent conditions at the start of the GFD, but rather novel conditions appearing favoured by the GFD itself (reference 6-9).

  • Body weigh increased > 10 kg was one of the criteria of metabolic alterations induced by GFD. However, higher body mass is associated with positive energy balance and general overconsumption, not necessarily related to GFD. Increased body weight could be also related to age of patients.

Author response: We are aware that the existing criteria defining metabolic alterations appearing after the start of the GFD has some intrinsic limitations. Still, any alteration which was not present at the start of the GFD and which appeared thereafter is usually considered as GFD-related, provided that it appears within a reasonable amount of time. In most studies, metabolic alterations appear within the first 12-24 months after the start of the GFD in the majority of patients. Our findings largely confirm these data (Figure 3). As such, we shared the common interpretation of metabolic alterations induced by to a positive energy balance due to an unfavorable nutritional contents of the gluten-free foods (reference 6-9).

  • Please provide explanation how was the compliance to GFD measured. Please provide the information about the methods of the analysis of the quality of patients’ diet. In the methodology there is only a statement (line 135): absence of accidental or intentional gluten ingestions, and in the result section there is only a short information (line 191) The vast majority of the patients were correctly compliant to the GFD (n=468, 87.0%). Please provide explanation who, when and using what methodology assessed the compliance to the GFD.

Author response: Actually, four different criteria were adopted to analyses adherence, hot merely absence of accidental or intentional gluten ingestions, but also remission of all CD-related symptoms; 3) normal tTGA levels; 4) Biagi score>2 (please refer to paragraph 2.4). All of these parameters are widely accepted criteria to assess compliance to the GFD, as a gold-standard is absent. The assessments were performed during the evaluations according to the same methodology reported in the referenced papers.

  • To follow GFD patients have to be educated about the principles of the diet. Please explain how patients were educated and who educated the patients. Was education similar for younger and older patients?  

Author response: We added the following specification: “Patients were systematically educated about GFD at the diagnosis and at each follow-up visit by the physicians of the clinics. In particular, we provided information about which foods contains gluten, which might be contaminated by gluten, and which were safe and permitted. This educational process was similar for younger and older patients. All of the physicians had a minimum 5-year expertise in the management of CD and GFD.”

  • Please explain prom the patient point of view your statement (lines 263-264).. that a follow-up shorter than two years might not be in the best interest of the patients.

Author response: We added the following specification: “a follow-up shorter than two years might not be in the best interest of the patients as they would be deprived of a medical support at a time in which problems related either to the CD or GFD are more likely to appear”.

  • Please explain in more detail the sentence (lines 277-278): While no elements suggest that the behavior of patients and physicians is different across our Country, caution should be adopted in interpreting our results. How did the authors measure the behavior of the patients?

Author response: We are sorry if we used the term “behavior” in a way which not completely appropriate for this context. We modified our sentence to avoid any possible misleading message: “While no elements suggest that the adherence of patients or the education provided by the physicians are different across our Country, caution should be adopted in interpreting our results”

  • Please explain which study findings are related to the conclusion provided in the lines 300-303 and 304-306.

Author response:  We modified the text as follows:

“In conclusion, we reported for the first time that the adherence to the follow-up procedures is heterogeneous (with patients at increased risk of complications being less likely to drop out from the follow-up visits)”

“…adopting different follow-up lengths based on the risk of complicated CD is actually a fact in everyday clinical practice, which is driven by the patients’ adherence rather than by medical prescriptions.”

  • Please give your opinion about the model of the follow-up procedures at 0, 1 month and 18-24 months after CD diagnosis and the time for each visit. Is it an optimal model for education and an evaluation of the results? Is always the problem related to patients? Should health care personnel be evaluated for their effectiveness?

Author response: We are grateful to the reviewer for this suggestion. We added the following considerations: “While the current guidelines suggestion of visits at 0, 6, and 18-24 months envision a good model for education and evalaution of results, their actual application remain an open problem, with healthcare professional needing to continously motivate patients toward the follow-up procedures (and not merely prescribing controls)”. 

  • I suggest reconsidering your conclusions to ensure they are consistent with the purpose and findings of the study.

Author response: Please refer to the responses from the previous two points.

Round 2

Reviewer 1 Report

The authors have satisfactorily responded to all the comments and imporved the manuscript accordingly. 
In have only one minor comment concerning the sentence "absence of referred accidental or intentional gluten ingestions". Do you possibly mean "absence of self-reported accidental or intentional gluten ingestions"? Please clarrify the use of this term. 

Reviewer 2 Report

I would like to thank the authors for their responses. I accept these explanations. However, I think that a major limitation of the study is the lack of an evaluation of diet quality in terms of gluten-containing products.